# Metoprolol Inhibits Developmental Brain Sterol Biosynthesis in Mice

**DOI:** 10.3390/biom12091211

**Published:** 2022-08-31

**Authors:** Luke B. Allen, Károly Mirnics

**Affiliations:** 1Munroe-Meyer Institute for Genetics and Rehabilitation, University of Nebraska Medical Center, Omaha, NE 68105, USA; 2Department of Biochemistry and Molecular Biology, College of Medicine, University of Nebraska Medical Center, Omaha, NE 68198, USA; 3Department of Pharmacology and Experimental Neuroscience, College of Medicine, University of Nebraska Medical Center, Omaha, NE 68198, USA; 4Department of Psychiatry, College of Medicine, University of Nebraska Medical Center, Omaha, NE 68198, USA

**Keywords:** beta-blockers, LC-MS/MS, 7-DHC, DHCR7, desmosterol, cholesterol, nebivolol, propranolol, atenolol, cell culture, neuron, astrocyte

## Abstract

De novo sterol synthesis is a critical homeostatic mechanism in the brain that begins during early embryonic development and continues throughout life. Multiple medications have sterol-biosynthesis-inhibiting side effects, with potentially detrimental effects on brain health. Using LC-MS/MS, we investigated the effects of six commonly used beta-blockers on brain sterol biosynthesis in vitro using cell lines. Two beta-blockers, metoprolol (MTP) and nebivolol, showed extreme elevations of the highly oxidizable cholesterol precursor 7-dehydrocholesterol (7-DHC) in vitro across multiple cell lines. We followed up on the MTP findings using a maternal exposure model in mice. We found that 7-DHC was significantly elevated in all maternal brain regions analyzed as well as in the heart, liver and brain of the maternally exposed offspring. Since DHCR7-inhibiting/7-DHC elevating compounds can be considered teratogens, these findings suggest that MTP utilization during pregnancy might be detrimental for the development of offspring, and alternative beta-blockers should be considered.

## 1. Introduction

Cholesterol is an essential molecule of life and is particularly important for brain development and function [1]. Although the human brain only accounts for about 2% of the total body weight, it contains as much as 25% of the cholesterol and cholesterol derivatives in the body [2,3,4]. The developing brain begins synthesizing its own cholesterol during embryonic development [3,4,5,6]. Cholesterol in the central nervous system (CNS) functions as much more than a structural component of cellular membranes and lipid rafts: it is required for synapse and dendrite formation as well as axonal guidance, and it serves as a precursor for various biosynthetic pathways [7,8,9].

The last step in the Kandutsch–Russell arm of the cholesterol biosynthesis pathway is the conversion of 7-dehydrocholesterol (7-DHC) to cholesterol, catalyzed by a single enzyme, 7-dehyrocholesterol reductase (DHCR7) (Appendix A) [10]. DHCR7 inhibition leads to reduced levels of cholesterol and desmosterol (DES)—cholesterol’s precursor in the Bloch sterol biosynthesis pathway—and accumulation of their immediate precursors, 7-DHC and 7-dehyrodesmosterol (7-DHD) [11,12,13,14]. 7-DHC and 7-DHD are the most reactive lipids known to date with a propagation rate constant of 2160 (this is 200-times more than cholesterol and 10-times more than arachidonic acid) [11].

This results in the formation of highly reactive autoxidation sterols, called 7-DHC-derived oxysterols [15,16,17]. These 7-DHC-derived oxysterols are reactive electrophiles that impair cell viability, differentiation, and growth [18,19,20]. They are not only markers of oxidative stress but are also biologically potent compounds capable of impeding or altering immune function [21]. In addition, DHCR7 inhibitors, which elevate 7-DHC levels and reduce DES levels during pregnancy are considered teratogens [22]. Finally, two mutant copies of the DHCR7 gene are either lethal (if sterol synthesis is fully eliminated) or result in a severe developmental disability called Smith-Lemli-Opitz syndrome (if some amount of residual cholesterol synthesis is preserved) [23,24,25].

As the blood–brain barrier (BBB) prevents passage of cholesterol from the rest of body to the central nervous system, the brain fully relies on its own cholesterol biosynthesis [6]. However, a significant number of medications with sterol biosynthesis inhibiting side effects can and do cross BBB—including haloperidol, aripiprazole, cariprazine, buspirone, trazodone, and others [26,27,28,29,30,31,32,33]. Our preliminary, high-throughput screening data on Neuro2a cells data suggested that commonly used beta-blockers for treatment of cardiovascular disease that pass through the BBB may interfere with normal sterol biosynthesis through off-target effects on the DHCR7 enzyme [34,35,36].

Chronic and gestational hypertension are the most common disorders of pregnancy and can result in premature birth [37]. Both are treated in order to prolong pregnancy and maximize the gestational age of the infant [38]. At the same time, maternal exposure to medications must be considered to avoid potential adverse effects on the developing fetus. The utilization of antihypertensive drugs during pregnancy, and the potential effect on the developing fetus have been topics of considerable debate: some studies found no negative effect on fetal development [39,40], while others reported increased risk for intrauterine growth restriction, small for gestational age, and preterm delivery [41,42,43]. As a result, we embarked on assessing the effects of six commonly used beta-blockers on sterol biosynthesis, utilizing cell lines, neuronal and astroglial cultures, and an in vivo mouse model.

## 2. Materials and Methods

### 2.1. Chemicals

Unless otherwise noted, all chemicals were purchased from Sigma-Aldrich Co (St. Louis, MO, USA). HPLC-grade solvents were purchased from Thermo Fisher Scientific Inc. (Waltham, MA, USA). All sterol standards, natural and isotopically labeled, used in this study are available from Kerafast, Inc. (Boston, MA, USA).

### 2.2. Cell Line Cultures

Human hepatocellular carcinoma HepG2 cells and the mouse neuroblastoma cell line Neuro2a, were purchased from ATCC (Rockville, MD, USA). The human and mouse cell lines were maintained in EMEM supplemented with L-glutamine, 10% FBS, and puromycin at 37 °C and 5% CO_2_. Cells were subcultured once a week and the culture medium changed every 2 days. For experimental purposes, the cells were plated in 96-well plates (for cell viability and sterol analysis).

To assess the endogenous sterol synthesis, these cultures were grown in defined medium without cholesterol and without lipids by using EMEM with N2 supplement and L-glutamine. At the endpoint of the incubation, Hoechst dye was added to all wells in the 96-well plate and the total number of cells counted using an ImageXpress Pico and cell counting algorithm in CellReporterXpress. After removing the medium, wells were rinsed twice with 1× PBS and then stored at −80 °C for lipid analysis. All samples were analyzed within 2 weeks of freezing.

### 2.3. Primary Neuronal Cultures

Primary cortical neuronal cultures were prepared from E15 and E17 mice as previously described [19,44,45]. Briefly, the brain was placed in pre-chilled HBSS solution (without Ca^2+^ or Mg^2+^), and two cortices were dissected and cut with scissors into small chunks of similar sizes and transferred to Trypsin/EDTA (0.5%) for 25 min at 37 °C. Trypsin was removed and residual trypsin inactivated by adding Trypsin Inhibitor (Sigma T6522) for 5 min. Solution was removed, and small tissue chunks were resuspended in Neurobasal medium (NBM) with B-27 supplement (Gibco #17504–044).

Samples were then spun at 80× *g*, the pelleted tissue was resuspended in NBM with B-27 supplement and then triturated with a fire-polished Pasteur pipette. The cells were pelleted by centrifugation for 5 min at 80× *g*. The cell pellet was resuspended in NBM with B-27 supplement and the cells were counted. The cells were plated on poly-D-lysine coated 96-well plates at 60,000 cells/well. The growth medium was NBM with B-27 supplement, Glutamax and 3 μM cytosine arabinoside. Cells were incubated at 37 °C, 5% CO_2_ for 6 days.

At the endpoint of the incubation, Hoechst dye was added to all wells in the 96-well plate, and the total number of cells counted using an ImageXpress Pico and cell counting algorithm in CellReporterXpress. After removing the medium, wells were rinsed twice with 1× PBS and then stored at −80 °C for sterol analysis. All samples were analyzed within 2 weeks of freezing. For the current study we prepared three independent preparations of primary cortical neurons from E15 (ATN, CRV, LAB, PRO) and E17 (MTP, NEB). Previous studies by our group have shown that different embryonic preparations do not respond differently to drugs other than inherent difference in the absolute level of sterols [45]. The results were concordant for cultures obtained from both embryonic stages.

### 2.4. Primary Astroglia Cultures

After plating the required number of cells for neuronal cultures, left-over cells were plated in 100 mm dishes at density of 1 × 10^7^ per tissue culture plate in DMEM with 10% FBS. Under these conditions, astrocytes adhere and divide and completely populate the plate within 10–14 days. Once the plates were full, they were rinsed using the cold jet method [46]. The astrocytes were trypsinized and plated in 96-well plates in DMEM plus 10% FBS at 30,000 cells/well. The following day, the medium was completely changed, and astrocytes were grown in Neurobasal medium with B-27 supplement in the absence of cholesterol (same medium as neuronal cells without cytosine arabinoside). Cells were incubated at 37 °C, 5% CO_2_ for 6 days. At the endpoint of the incubation, cells were counted as described for neuronal cultures and processed in a similar manner for the analysis as neuronal cultures.

### 2.5. Mouse Studies

Adult male and female C57BL/6J stock # 000.664 mice were purchased from Jackson Laboratories. Mice were housed under a 12 h:12 h light-dark cycle at constant temperature (25 °C) and humidity with ad libitum access to food (Teklad LM-485 Mouse/Rat Irradiated Diet 7912) and water in Comparative Medicine at the UNMC, Omaha, NE. The time-pregnant female mice received IP injections of vehicle (VEH) or MTP (35 mg/kg) from E12 to E19. The mice were monitored daily during injections and E19 offspring were collected at least 2 h after the final injection of either VEH or MTP.

Adult female mice were also sacrificed at the same time as pups. Frozen brain tissue samples were sonicated in ice-cold PBS containing butylated hydroxytoluene (BHT) and triphenylphosphine (PPh_3_). The aliquots of homogenized tissue were used for sterol extraction and protein measurements. The protein was measured using Pierce’s BCA assay. All procedures were performed in accordance with the Guide for the Humane Use and Care of Laboratory Animals. The use of mice in this study was approved by the Institutional Animal Care and Use Committee of UNMC.

### 2.6. Cell Sterol Measurements

Sterol levels were analyzed in individual wells of 96-well plate, and, for most experiments, cellular levels correspond to 4–8 technical replicates. After rinsing plates with 1× PBS (or removing previously frozen plates from −80 °C freezer) 200 μL of MeOH containing the internal standard cocktail was added, as reported previously [45,47]. The plate was covered by aluminum foil and then placed on an orbital shaker for 30 min at room temperature. An aliquot (100 µL) of the supernatant was transferred to a plate predeposited with 200 µg of 4-phenyl-1,2,4-triazoline-3,5-dione (PTAD) and sealed with Easy Pierce Heat Sealing Foil followed by 30 min agitation at room temperature and analyzed by LC-MS/MS (as described in a following section). The values were normalized by cell count for each well and are reported as the fold change over control (vehicle, DMSO) or nmol/million cells.

### 2.7. Mouse Sterol Measurements

To 10 µL of mouse serum or tissue lysate, we added 800 µL of Folch solution containing 0.25 mg/mL TPP, 0.005% BHT, and the internal standards d_7_-7-DHC (13 ng), ^13^C_3_-DES (100 ng), ^13^C_3_-LAN (100 ng), d_7_-CHOL (34 ng), followed by the addition of 400 µL of 0.9% NaCl. The resulting mixture was vortexed and centrifuged. The lower organic phase was recovered and dried in SpeedVac. 100 µL of 2 mg/mL freshly prepared PTAD solution in MeOH was added to the residues of the extracts for derivatization as previously described [47].

The solutions were incubated for 30 min at room temperature with constant shaking, transferred into sample vials and placed in an Acquity UPLC system equipped with ANSI-compliant well plate holder coupled to a Thermo Scientific TSQ Quantis mass spectrometer equipped with an APCI source. Then, 10 μL was injected onto the column (Phenomenex Luna Omega C18, 1.6 μm, 100 Å, 2.1 × 100 mm) with 60% MeOH (0.1% *v*/*v* acetic acid) 40% acetonitrile (0.1% *v*/*v* acetic acid) mobile phase for a 1.4 min runtime at a flow rate of 500 μL/min. 

Natural sterols were analyzed by selective reaction monitoring (SRM) using the following transitions: Chol 369 → 369, 7-DHC 560 → 365, desmosterol 592 → 560, and lanosterol 634 → 602, with retention times of 1.5, 0.78, 0.55, 0.76, and 0.61 min, respectively. SRMs for the internal standards were set to: d_7_-CHOL 376 → 376, d_7–_7-DHC 567 → 372, ^13^C_3_-DES 595 → 563, and ^13^C_3_-LAN 637 → 605. Final sterol numbers are reported as nmol/µL of serum or nmol/mg protein.

### 2.8. Drug Extraction

Drug extraction was performed using our previously published protocol [48,49]. Briefly, to 100 µL of sample, we added 188 µL H_2_O, 10 µL internal standard (d_8_-aripiprazole) and 75 µL of 4.5 M ammonium hydroxide. The samples were then vortexed vigorously for 1 min. Following vortexing, 950 µL of methyl *tert*-butyl ether (MTBE) was added to each sample followed by another minute of vortexing. Samples were then centrifuged at 10,000× *g* for 10 min. The top organic layer was collected to a glass autosampler vial and dried in a SpeedVac. After samples were dried 100 µL of MeOH/NH_3_·H_2_O (95:5, *v*/*v*) was added to each vial, briefly vortexed and then transferred to a chromatography vial insert for LC-MS/MS analysis.

### 2.9. Drug Measurements

Metoprolol levels were acquired in an Acquity UPLC system coupled to a Thermo Scientific TSQ Quantis mass spectrometer using an ESI source in the positive ion mode. 10 μL of each sample was injected onto the column (Phenomenex Luna Omega C18, 1.6 μm, 100 Å, 2.1 × 50 mm) using water (0.1% *v*/*v* acetic acid) (solvent A) and acetonitrile (0.1% *v*/*v* acetic acid) (solvent B) as mobile phase. The gradient was: 10% to 40% B for 0.5 min; 40% to 95% B for 0.4 min; 95% B for 1.5 min; 95% to 10% B for 0.1 min; 10% B for 0.5 min. Metoprolol was analyzed by selective reaction monitoring (SRM) using the following transition: metoprolol 268 → 133. The SRM for the internal standards (d_8_-aripiprazole) was set to 456 → 293, and the response factors were determined to accurately determine the drug levels. The final drug levels are reported as ng/mg of protein.

### 2.10. Statistics

Statistical analyses were performed using GraphPad Prism 9 for Windows. Unpaired two-tailed *t*-tests were performed for individual comparisons between two groups. The Welch’s correction was employed when the variances between the two groups were significantly different. Ordinary one-way ANOVA with Dunnett’s multiple comparison correction was employed to compare the difference between each concentration and the control condition. When the Brown–Forsythe test for equal variances returned *p* < 0.05 Brown–Forsythe and Welch ANOVA tests were employed with Dunnett’s T3 correction for multiple comparisons. The *p* values for statistically significant differences are highlighted in the figure legends.

## 3. Results

### 3.1. Multiple Beta-Blockers Inhibit Sterol Biosynthesis In Vitro

To investigate the effects of commonly used beta-blockers on sterol biosynthesis, we selected six commonly used beta-blockers that were prescribed > 125 million times/year in the US alone (Appendix A) [50]. The screening for sterol biosynthesis inhibiting properties of atenolol (ATN), carvedilol (CRV), labetalol (LAB), metoprolol (MTP), nebivolol (NEB), and propranolol (PRO) was performed in four distinct in vitro systems: Neuro2a and HepG2 cell lines, primary neurons and astrocytes. We tested the medications at 100 nM, 200 nM, 500 nM, and 1 µM concentrations and compared the outcomes to vehicle-exposed cultures.

Across all four experimental systems we observed a sharp rise in 7-DHC and DES reduction for NEB, MTP, and LAB (Figure 1). This effect was most prominent for all medications in HepG2 cells, while the weakest response was observed in Neuro2a cells. The most potent inhibitor of sterol biosynthesis was NEB (up to 600-fold 7-DHC increase in HepG2 cells), followed by MTP (200-fold 7-DHC increase in HepG2 cells). The observed effects were dose-dependent for NEB, MTP, and LAB, with no significant effects of ATN, CRV and PRO on sterol biosynthesis (Appendix A). Notably, while both NEB and MTP significantly reduced CHOL levels, this reduction was small in comparison to the observed DES reduction. Lanosterol (LAN) levels were not significantly changed in any of our experiments as the LAN synthesizing enzymes are not affected by any of the tested medications.

### 3.2. MTP Is a Strong Inhibitor of Sterol Biosynthesis In Vitro

While NEB showed the highest sterol biosynthesis inhibitory effects in our initial screening, based on the sheer volume prescription we decided to focus our follow-up experiments on MTP. MTP is the fifth most commonly utilized medication in the US, with >75 M yearly prescriptions [50]. It is often prescribed to pregnant women [37,39,40,41,42,48,51,52,53,54]. In the preliminary experiments MTP exposure resulted in a 50–100-fold elevation of 7-DHC in neurons and astrocytes. These increased 7-DHC levels were concerning, as patients with Smith–Lemli–Opitz syndrome (SLOS), due to compound genetic mutations of the DHCR7 gene show a 30–50-fold elevation of 7-DHC in their sera compared to sera of healthy individuals. However, we acknowledge that the magnitude of changes in our in vitro system cannot be directly compared to the 7-DHC elevation seen in SLOS patients.

The dose-dependent effects of MTP on post-lanosterol synthesis are shown in Figure 2 and Figure 3. In comparison to vehicle-treated cultures, the strongest, highly significant increase were observed in 7-DHC levels. The data across the 4 cell lines suggested that the degree of sterol biosynthesis inhibition by MTP is highly dependent on the tissue type: HepG2 cells were approximately 50-fold more affected by MTP than Neuro2a cells (200-fold increase over vehicle-treated baseline vs. five-fold change over baseline for 7-DHC levels; Figure 2).

Due to the apparent DHCR7 inhibition by MTP both the Kandutsch–Russel and Bloch branches of the sterol biosynthesis were affected (Appendix A), and we also observed a strong and significant decrease in DES. Residual DES levels decreased to approximately 45% of those seen in vehicle-treated HepG2 cultures, with the smallest effects observed in Neuro2a cells (70% of sham-treated cultures). CHOL levels seemed to be mostly resilient to MTP treatment across all cell lines (except the highest concentration in HepG2 cells), which is likely due to the short treatment window, residual CHOL level in the cultures, and slow CHOL turnover.

### 3.3. MTP Inhibits Sterol Biosynthesis in Maternal Mouse Tissues

Due to the well-documented biological effects of 7-DHC on development, obtained in vitro data, and the use of MTP by pregnant women, we decided to test the MTP effects in a maternal exposure model in mice. Pregnant mice were injected daily with 35 mg/kg MTP or vehicle from embryonic day 12–19 (E12–E19). This dose was selected to mimic the dosage typically prescribed to human patients.

At E19, the pregnant dams were killed, and the maternal brain, heart, kidney, liver, sciatic nerve, and serum were harvested for sterol profiling. Maternal brain regions were further dissected, and the neocortex, hippocampus, striatum, midbrain, cerebellum, and medulla/pons were also analyzed using LC-MS/MS. The analysis of brain regions revealed that all six investigated maternal brain regions had significantly elevated 7-DHC levels (Figure 4). In addition, we also observed a relatively uniform, significant 7-DHC elevation in the heart, kidney, sciatic nerve, and serum of the same pregnant dams (Appendix A). Presumably due the low turnover rate of CHOL and short treatment window, we did not observe significant changes in CHOL levels in any of the investigated tissues (Appendix A).

### 3.4. Maternal Exposure to MTP Alters the Sterol Composition in the Tissue of Offspring

Next, we assessed if maternal MTP can cross the placenta and impact embryonic brain sterols biosynthesis. Maternal exposure to MTP resulted in a strong and highly significant, >10-fold elevation of 7-DHC in the offspring brain (*p* < 0.0001) (Figure 5). In contrast, DES levels decreased by approximately 15% (*p* < 0.001), with non-significant effects on CHOL and LAN. As a result, 7-DHC/CHOL ratio showed an approximately nine-fold increase (*p* < 0.0001). We did not observe any differences in the sterol profile between male and female pups in response to the maternal MTP treatment.

Following the assessment of the brain, we performed a similar assessment of post-lanosterol sterol biosynthesis on the heart (Figure 6) and liver (Figure 7) of offspring that were maternally exposed to MTP. The overall results in both peripheral tissues were comparable to those that seen in the brain, suggesting that MTP action is consistent on the peripheral and central sterol biosynthesis. Notably, absolute analyte levels were different across the three investigated tissues, with the CNS pool showing the highest absolute sterol levels at baseline (CHOL: Brain—181 nM/mg protein; Heart—73 nM/mg protein; Liver—70 nM/mg protein).

These results suggest that MTP is a strong developmental inhibitor of sterol biosynthesis across both the CNS and peripheral tissues, primally interfering with the function of the DHCR7 enzyme. Notably, no statistically significant sex differences were observed.

### 3.5. Correlation of MTP Levels with Sterol Precursors

Finally, we examined the correlation between DES and 7-DHC levels with MTP levels in the offspring brain and liver (Figure 8). DES levels were not correlated with the observed MTP levels in the exposed offspring brain or liver, while a complex correlation pattern emerged for the 7-DHC/MTP levels. In the brain, 7-DHC showed a significant, negative correlation with MTP levels (R^2^ = 0.32, *p* = 0.0001), while in the liver 7-DHC showed a strong, positive correlation with MTP (R^2^ = 0.78, *p* < 0.0001).

We attribute this to the very short half-life of MTP in the brain, and a delay between maximum brain concentration of MTP and peak 7-DHC concentration. We propose that while MTP levels are the highest in the brain, the 7-DHC levels are still rising, and by the time 7-DHC levels peak MTP is already cleared from the brain tissue. Apparently, this fast turnover of MTP is not present in the liver of the same maternally exposed offspring, as the 7-DHC and MTP levels are almost perfectly correlated.

## 4. Discussion

The six beta-blockers we tested accounted for over 125 million prescriptions in 2019 in the US alone [55]. While beta-blockers are generally well-tolerated and are known to be safe and life-saving drugs, they have also been known to cause CNS side effects in a subset of patients [56,57,58,59,60,61,62,63,64]. These side effects include but are not limited to psychiatric conditions, such as bizarre and vivid dreams, sleep disturbances, delirium, psychosis and visual hallucinations [60,64]. These CNS effects have been primarily correlated with the lipophilicity of the beta-blocker, which allows for rapid penetration of the blood-brain barrier [64,65,66,67]. 

Furthermore, recent studies have revealed that chronic administration of MTP induced markers of phagocytosis in the brain and impaired cognitive behavior in both wild-type and amyloid-beta protein precursor model of Alzheimer’s disease in mice [68].

On the other hand, hydrophilic beta-blockers, such as atenolol have been shown to induce signaling molecules, such as NO and H_2_O_2_ in the hypothalamus without crossing the BBB [69]. Hydrogen peroxide can contribute to oxidative stress [70], which can also affect sterol biosynthesis [17,71] and cell viability through the formation of oxysterols [18,19,20]. Additionally, oxysterols have been shown to play a role in CNS autoimmune processes [72]. This suggests that, while hydrophilic beta-blockers may not cross the BBB, they still may affect CNS development and function.

CNS-specific side effects of beta-blockers have been assumed to be a result of adrenergic receptor-mediated mechanisms [73]. However, in light of our findings, we believe that the explanation is more complex. We propose that MTP side effects arise as a complex interplay between beta-adrenergic effects [62,74] and sterol biosynthesis disruptions [75]. Thus, deciphering the root cause of the observed side effects stands as a difficult task, further complicated by seemingly multiple mechanisms of action occurring simultaneously [76,77]. This remains an interesting avenue for future studies, particularly when considering the large number of beta-blocker prescriptions currently being given.

Our studies show that the developing brain is extremely susceptible to the sterol biosynthesis inhibiting effects of MTP, and perhaps other commonly used beta-blockers, such as nebivolol. Worryingly, beta-blockers are the most prescribed cardiac medications for pregnant women [78]. Furthermore, psychotropic medications with sterol biosynthesis side effects (aripiprazole, trazodone, haloperidol, and cariprazine) [1,48] are also commonly prescribed during pregnancy, often by different providers.

This raises the possibility that the sterol biosynthesis inhibiting effects of these two classes of medications will summate, further elevating the 7-DHC levels to dangerous levels in the developing brain. This view is supported by a recent study showing that long-term exposure to MTP as a part of polypharmacy (concurrent use of five or more drugs) lead to a deficit in cognitive function in young adult female mice [79].

The potential interactions between beta-blockers and psychotropic medications are also important to consider in adult patients, not only during pregnancy. There is a high level of comorbidity between cardiovascular diseases which may call for beta-blockers and bipolar disorder or schizophrenia [80]. In fact, beta-blockers accounted for 25% of all cases that had interactions with antipsychotic medications [80]. Unfortunately, while it is considered a questionable practice, beta-blockers are often prescribed alongside standard antipsychotic treatments to treat cardiovascular disease and extrapyramidal symptoms, such as akathisia [81,82,83].

One should also consider the potential for interactions as a result of sterol-inhibiting polypharmacy, as it appears that there are multiple chemical substructures that inhibit DHCR7. We previously identified 2,3-dichlorophenylpiperazine (2,3-DCPP) as being capable of inhibiting DHCR7 [84]. However, unlike aripiprazole and cariprazine, two DHCR7 inhibitors sharing the 2,3-DCPP motif [30,31,45,48,84,85,86,87] the beta-blockers discussed in the present study contain no such common moieties (Appendix A), suggesting that there are multiple chemical substructures that may inhibit DHCR7 and elevate 7-DHC.

The public health implications of our findings are noteworthy. As obesity is one of the leading risk factors for high blood pressure and other cardiovascular disease [88,89,90,91,92,93], the ongoing obesity pandemic [92,94,95] would suggest that MTP prescriptions will continue to increase. This will likely extend to pregnant women as well, as MTP is not currently contraindicated for pregnancy. Thus, it is likely that prescription of MTP (and other beta-blockers) will continue to increase in pregnant women to control blood pressure due to obesity, thereby, exposing more fetuses to a potentially harmful increase in 7-DHC during critical developmental periods.

This may be further compounded in the 1–3% of the human population that carry single-allele DHCR7 mutations who already have increased baseline 7-DHC levels [13,96,97] where the potential for a drug × gene interaction could lead to even worse outcomes. In summary, based on the documented teratogenicity of DCHR7 inhibitors [22], the DHCR7 inhibitory side effects of MTP are a cause for caution, particularly in the context of pregnancy, polypharmacy and DHCR7 genetic liability.

## Figures and Tables

**Figure 1 biomolecules-12-01211-f001:**
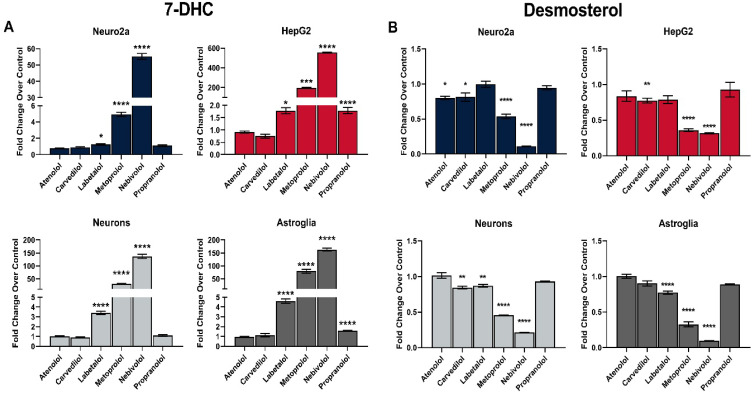
Select β-blockers elevate 7-DHC (**A**) and reduce DES (**B**) levels in vitro. Neuro2a, HepG2, primary neurons and astroglia were treated with 1 µM of six β-blockers or vehicle in cholesterol-free defined media for 48 h (Neuro2a and HepG2) or 6 days (neurons and astroglia). 7-DHC levels were analyzed by LC-MS/MS. The y-axes have different scales for visualization purposes. Note that NEB, MTP, and LAB robustly and significantly increase 7-DHC in all cell types. Values reported as mean ± SEM over vehicle-treated control; statistical significance: * *p* < 0.05; ** *p* < 0.01 *** *p* < 0.001; **** *p* < 0.0001.

**Figure 2 biomolecules-12-01211-f002:**
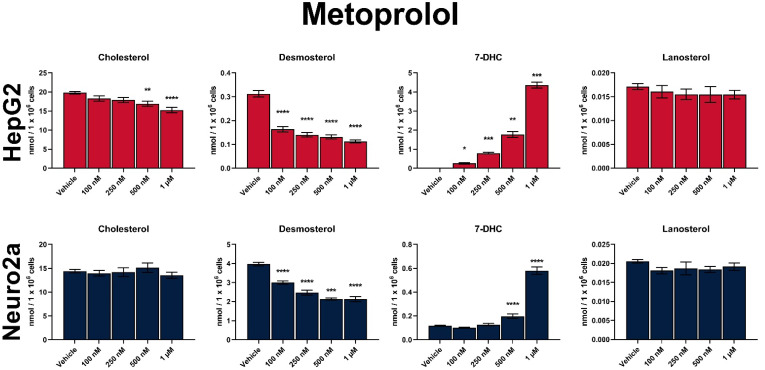
Dose-dependent sterol response of HepG2 and Neuro2a cells treated with MTP. Values reported as nmol/million cells. Statistical significance: * *p* < 0.05; ** *p* < 0.01; *** *p* < 0.001; **** *p* < 0.0001 vs. vehicle treatment.

**Figure 3 biomolecules-12-01211-f003:**
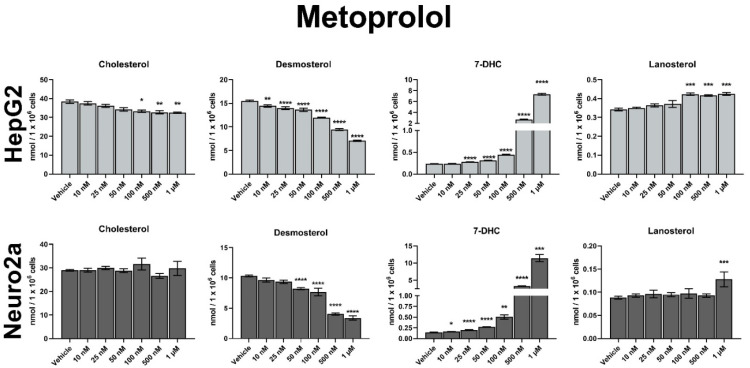
Dose-dependent sterol response of neurons and astrocytes treated with MTP. Values reported as nmol/million cells. Note the broken *y*-axis for 7-DHC. Statistical significance: * *p* < 0.05; ** *p* < 0.01; *** *p* < 0.001; **** *p* < 0.0001 vs. vehicle treatment.

**Figure 4 biomolecules-12-01211-f004:**
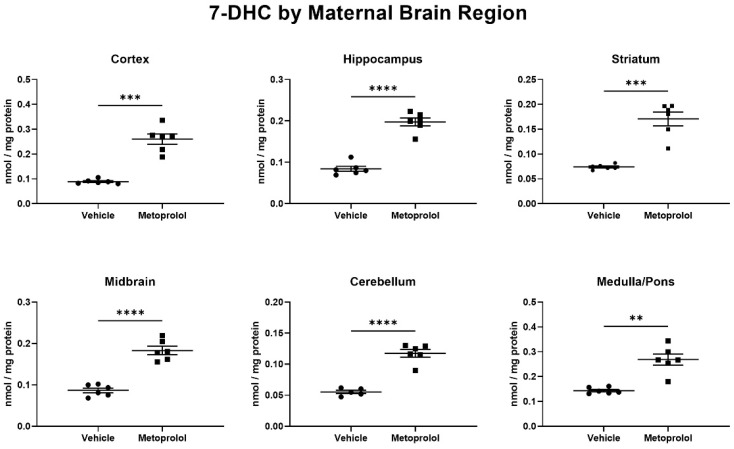
Metoprolol elevates 7-DHC throughout the maternal brain. Pregnant mice were treated with 35 mg/kg MTP from E12 to E19. Dam brains were analyzed for sterol content by LC-MS/MS. Statistical significance: ** *p* < 0.01; *** *p* < 0.001; **** *p* < 0.0001 vs. vehicle treatment. Note that 7-DHC baseline levels in the vehicle-treated animals are different across the brain regions, and that a robust 7-DHC elevation is evident in all investigated brain regions. In the same brain regions, cholesterol levels were unchanged (Appendix A).

**Figure 5 biomolecules-12-01211-f005:**
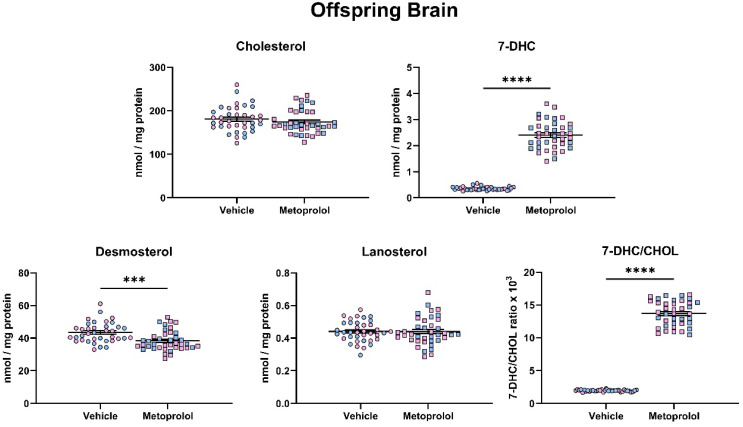
Maternal exposure to metoprolol results in strong elevation of 7-DHC and decreased desmosterol levels in the offspring brain. Pregnant mice were treated with 35 mg/kg MTP from E12 to E19. E19 brains were analyzed for sterol content by LC-MS/MS. Each symbol represents a sample originating from a single offspring brain. Blue fill = male embryos, pink fill = female embryos. Note the strong change in 7-DHC/CHOL (precursor/product) ratio. Statistical significance: *** *p* < 0.001; **** *p* < 0.0001.

**Figure 6 biomolecules-12-01211-f006:**
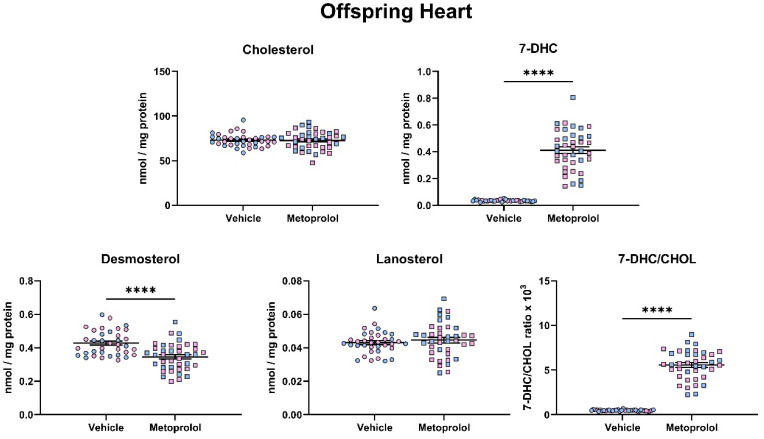
Maternal exposure to metoprolol results in strong elevation of 7-DHC and decreased desmosterol levels in the offspring heart. Pregnant mice were treated with 35 mg/kg MTP from E12 to E19. E19 hearts were analyzed for sterol content by LC-MS/MS. Each symbol represents a sample originating from a single offspring heart. Blue fill = male embryos, pink fill = female embryos. Note the strong change in 7-DHC/CHOL (precursor/product) ratio. Statistical significance: **** *p* < 0.0001.

**Figure 7 biomolecules-12-01211-f007:**
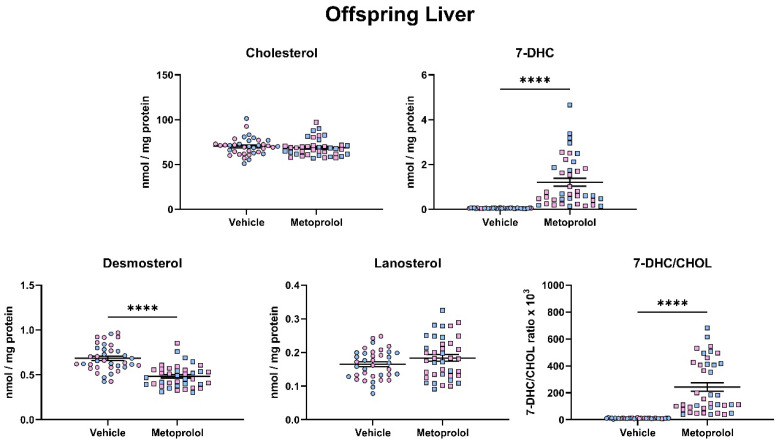
Maternal exposure to metoprolol results in strong elevation of 7-DHC and decreased desmosterol levels in the offspring liver. Pregnant mice were treated with 35 mg/kg MTP from E12 to E19. E19 livers were analyzed for sterol content by LC-MS/MS. Each symbol represents a sample originating from a single offspring liver. Blue fill = male embryos, pink fill = female embryos. Note the strong change in 7-DHC/CHOL (precursor/product) ratio. Statistical significance: **** *p* < 0.0001.

**Figure 8 biomolecules-12-01211-f008:**
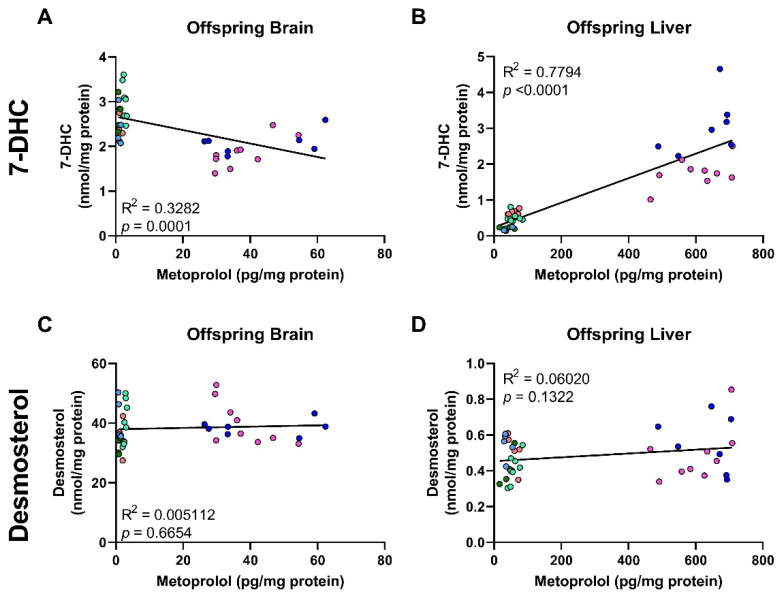
Metoprolol levels are correlated with 7-DHC levels (**A**,**B**) and DES levels (**C**,**D**) in the tissues of maternally exposed offspring. Only results obtained from MTP-exposed offspring are plotted. Each symbol corresponds to a single offspring sample, and samples from littermates are denoted by the same color. Lines denote simple linear regression. Note the variability between litters, but consistency within littermates.

## Data Availability

Not applicable.

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
