# Peer review of "Metoprolol Inhibits Developmental Brain Sterol Biosynthesis in Mice"

_biomolecules, 2022, doi:10.3390/biom12091211_

Round 1
Reviewer 1 Report
This is a fascinating study demonstrating that beta blockers can modulate the cholesterol biosynthesis pathway through inhibition of DHCR7. I only have only minor comments for the authors to consider.
1. Line 38, perhaps at this point it should be specified that 7-DHC to cholesterol is the last step in the KR pathway.
2. Line 149. How much PTAD is used per 100 microL supernatant.
3. It would help the reader if the PTAD derivatisations and MRM transitions were explained in the supplemental.
4. Line 168. I can't follow how 8-DHC and 8-DHD can have the same MRM transitions. Some explanation would help.
5. If the authors have a conversion factor between ng/mg protein and ng/g wet weight it could be useful to include for those more familiar with the latter unit.
Author Response
#1. Line 38 revised as requested
#2. Line 149 - amount specified
#3 PTAD derivatization description - This has been extensively described by the peer-reviewed publication "A highly sensitive method for analysis of 7-dehydrocholesterol for the study of Smith-Lemli-Opitz syndrome" that can be downloaded from PubMed central https://www.ncbi.nlm.nih.gov/pmc/articles/PMC3886672/. It would be redundant to basically copy the information from the manuscript. The reference is listed in the literature section.
#4. Line 168 - we eliminated 8-DHC, 7-DHD and 8-DHD data and discussion from the whole manuscript, as it is not a major, readily interpretable finding. A long technical, and convoluted explanation would only distract from the focus of the manuscript. The reviewer is correct in that 8-DHD would not be expected at this transition, however due to the natural distribution of isotopes ~8% of 8-DHD would be detected at this transition. Furthermore, as we do not have response factors or matched internal standards for 7-DHD and 8-DHD, we have excluded 8-DHC from the results. This revision does not affect for the rest of the findings as we have both matched internal standards and baseline separation for the other analytes.
#5 Conversion factor between ng/mg protein and ng/g wet weight - This is tricky, and very much variable across the experimental tissues, as it is also a function of water content. Furthermore, wet weight cannot be measured in cell culture experiments. We do nor feel comfortable speculating on this issue.
Reviewer 2 Report
This is a novel report that clearly demonstrates that commonly prescribed beta-blocker drugs can inhibit 7-dehydrocholesterol (7DHC) reductase (DHCR7), leading to abnormal accumulation of 7DHC and its oxysterol byproducts. Several of the latter compounds have been shown previously to be cytotoxic. The clinical relevance of this work primarily relates to the use of such drugs by pregnant women and the potential effects on the developing embryo/fetus. The study was nicely designed, well-crafted, and the data are compellingly presented. Both in vitro (cell culture) and in vivo (mouse) studies were performed, and the results correlate well. The manuscript is clearly written, and the conclusions drawn are reasonable.
I have no major concerns about any of it. However, I just have a few MINOR suggestions and comments for the authors to consider for making revisions:
1- I think it would be a nice addition to show (as a Supplemental figure) the structures of the beta-blocker drugs and to comment upon similarities or differences, particularly within the context of structure-function relationships (e.g., what features would lead one to predict these compounds to be potential inhibitors of DHCR7 or any other sterol pathway enzyme?).
2- Again considering chemical structure, how do these beta-blockers compare with known, relatively selective DHCR7 inhibitors (e.g., AY9944)? And comparing requisite molarity of each required to achieve some criterion % inhibition of DHCR7, how do these beta-blockers compare against, e.g., AY9944?
3- line 130: to be more precise, light-dark cycle would be better to indicate as 12 h:12 h light-dark cycle
4- line 149: no need to capitalize Phenyl
5- line 411- typo DCHR7 (should be DHCR7)
6- Throughout the manuscript, be consistent with abbreviations: just use h for hours, min for minutes. For microliters use (micron symbol)L (capital L, not lower case l)
Author Response
#1 - Chemical structures are added in supplemental material. We have been thinking about this for a long time, but it appears that there is not a single chemical substructure that is responsible for this inhibition across the different chemicals. In the case of ARI and CAR, the culprit is 2,3-DCPP (which we managed to prove previously), but we do not see similar substructures in beta-blockers or AY9944.
#2. Beta-blockers vs AY9944 - we have to be cautious comparing concentrations between the two chemicals. AY is not used clinically due to its toxicity and is a teratogen, and we are using MTP concentrations in mice in bioequivalent human doses. In cultures, AY9944 is definitively more potent than MTP, but these data often do not extrapolate to in vivo experiments.
#3-6 All formatting issues attended to as suggested by the reviewer.